# Low Molecular Weight Kappa-Carrageenan Based Microspheres for Enhancing Stability and Bioavailability of Tea Polyphenols

**Tao Feng** **, Kai Wu, Jianying Xu \*, Zhongshan Hu and Xiaolei Zhang**

School of Perfume and Aroma Technology, Shanghai Institute of Technology, Shanghai 201418, China; fengtao@sit.edu.cn (T.F.); 15738285855@163.com (K.W.); 18827511750@163.com (Z.H.); 15921669366@163.com (X.Z.)
\* Correspondence: soyjle_2000@163.com

**Abstract:** Tea polyphenols (TP) are a widely acknowledged bioactive natural product, however, low stability and bioavailability have restricted their application in many fields. To enhance the stability and bioavailability of TP under certain moderate conditions, encapsulation technique was applied. Kappa–Carrageenan (KCG) was initially degraded to a lower molecular weight KCG (LKCG) by $H_2O_2$, and was selected as wall material to coat TP. The obtained LKCG (Mn = 13,009.5) revealed narrow dispersed fragments (DPI = 1.14). FTIR and NMR results demonstrated that the main chemical structure of KCG remained unchanged after degradation. Subsequently, LK-CG and TP were mixed and homogenized to form LK-CG-TP microspheres. SEM images of the microspheres revealed a regular spherical shape and smooth surface with a mean diameter of 5–10 μM. TG and DSC analysis indicated that LK-CG-TP microspheres exhibited better thermal stability as compared to free TP. The release profile of LK-CG-TP in simulated gastric fluid (SGF) showed a slowly release capacity during the tested 180 min with the final release rate of 88.1% after digestion. Furthermore, in vitro DPPH radical scavenging experiments revealed that LK-CG-TP had an enhanced DPPH scavenging rate as compared to equal concentration of free TP. These results indicated that LK-CG-TP microspheres were feasible for protection and delivery of TP and might have extensive potential applications in other bioactive components.

**Keywords:** kappa-carrageenan; tea polyphenols; encapsulation; microspheres; sustainable release; DPPH scavenging

## 1. Introduction

Tea polyphenols (TP) are flavanols, flavandiols, flavonoids, and phenolic acids obtained from the leaves of tea (*Camelia sinensis*) plant, which have been recognized as the main bioactive compounds responsible for the beneficial effects and are now widely used in food, medical, pharmaceutical, and cosmetic fields [1,2]. TP have various heath promoting properties such as anti-oxidant, anti-cancer, anti-diabetic, anti-carcinogenic, anti-fungal, anti-viral, anti-obesity, and anti-inflammatory activities [3]. The molecules of TPs are mainly composed of four catechins, including (−)-epigallocatechin gallate (EGCG), (−)-epigallocatechin (EGC), (−)-epicatechin gallate (ECG), and (−)-epicatechin (EC) [4]. Among these catechins, EGCG is the most abundant compound in green tea, black tea, and oolong teas, and reveals a greater antioxidant effect due to the presence of its trihydroxy structure [2,5].

Numerous investigations have focused on the positive effects of TP on health benefits through in vivo and/or in vitro methods during the last several decades [6,7]. For example, oral TP could reveal anti-inflammatory and anti-obesity activities through regulating gut microbiota in canines [8]. However, the poor stability and low absorption rate of TP are great bottlenecks for their application in functional foods and other health beneficial products [2]. As reported elsewhere, TP are very unstable when they pass through salivary, gastric, and upper small intestinal phases of the digestion process [9]. Furthermore,

the unique environment of the gastrointestinal tract (i.e., pH, intestinal epithelium, enzymes, gastric juices etc.) could significantly affect the availability of TP [9]. In addition, TP are easily oxidized in the air, which would limit their bioactivities and subsequent applications [10].

In recent years, more and more attempts have been made at the study of delivery systems with controlled release of various bioactive components through encapsulation technology [11]. Meanwhile, many researchers have tried using this novel technology to improve the stability and bioavailability of TP to address its application drawbacks. During the encapsulation process, the selection of coating material is an important factor for protecting structural and functional stability of the core material. Up to the present, TP have been encapsulated in several formulations to strengthen their stability and bioavailability during oral and transdermal controlled delivery systems [12]. For TP's encapsulation, the most used coating materials are maltodextrins [13], arabic gum [14], gelatin [15], milk proteins [3], alginate, and chitosan [10], which have different advantages and disadvantages related to the stability, bioavailability, biocompatibility, and cost.

Carrageenan is a sulphated linear polysaccharide mainly derived from certain red seaweeds of the *Rhodophyceae* class, which consists of D-galactose residues linked alternately in 3-linked-β-D-galactopyranose and 4-linked-β-D-galactopyranose units [16]. The carrageenan polysaccharides are classified into six basic forms according to the degree of substitution that occurs on their free hydroxyl groups, i.e., kappa (κ)-, iota (τ)-, lambda (λ)-, mu (μ)-, nu (ν)-, and theta (θ)-carrageenan [17]. Carrageenan-based delivery systems present outstanding performance with regard to the delivery of bioactive molecules, which improved the stability and bioavailability of these compounds, and revealed its unique advantages compared with other encapsulation materials. For example, K-CG has a higher compatibility with acidic conditions than alginate, and resulted in better protection of encapsulated bioactive components [18]. Carrageenan holds promise for the food and pharmaceutical industry due to its superior properties, such as low cost, non-toxicity, biocompatibility, biodegradability, and low immunogenicity [19].

Carrageenan-based nutraceutical delivery systems have been extensively investigated recently. These systems mainly consist of nanoparticle-based delivery carriers, hydrogels, emulsions, complexes, microcapsules, microbeads, nanotubes, and aerogels [20]. For instance, K-CG was applied to form edible hydrophilic films with curcumin for the preservation of grass carp fillets due to its excellent film-forming ability and anti-oxidative property [21]. However, to date, the utilization of carrageenan for delivery of TP has seldom been investigated [22]. In this work, a delivery system was developed based on low molecular weight K-CG (LK-CG) for the first time, to enhance the stability and antioxidant activity of TP. The preparation of LK-CG and LK-CG-TP microspheres was optimized in the preliminary experiments, and characterization of LK-CG-TP was conducted through FT-IR, NMR, and SEM methods. Furthermore, stability, drug release, and antioxidant activity of LK-CG-TP were analyzed to evaluate properties and potential applications of these newly developed microspheres.

## 2. Materials and Methods

### 2.1. Material

K-CG was commercially obtained from Shanghai Brilliant Gum Co., Ltd. (Shanghai, China), with a Mass-average molecular weight (Mw) of $23 \times 10^4$ Da. TP powder was purchased from Shaanxi Sanyuan Senfu Biology and Technology Co., Ltd. (Xi'an, Shaanxi, China). Hydrogen peroxide, span 80 was commercially obtained from Titan Technology Co., Ltd. (Shanghai, China). *n*-butanol was purchased from National Medicine Group Chemical Reagent Co., Ltd. (Guangzhou, China). Dialysis bag was purchased from Solarbio Life Science Co., Ltd. (Beijing, China). Purified water was commercially obtained from Shenzhen Watsons Distilled Water Co., Ltd. (Shenzhen, China).

## 2.2. Preparation of LK-CG

Although carrageenans, including K-CG, have many biological functions, the high molecular weight and poor tissue-penetrating ability of the polysaccharides have largely limited their further utilization in both food and nonfood applications [23,24]. In the present work, a free radical depolymerization method was employed for preparation of LK-CG, with some modification [23]. K-CG was dissolved in 4% $H_2O_2$ solution (500 mL), and incubated at 80 °C for 4 h. Thereafter, the aqueous solution was transferred to a dialysis bag (7000 Da) and dialyzed in running deionized water for 6 days. Then, the dialyzed liquid was vacuum freeze-dried to obtain LK-CG.

The Mw and Mn of obtained LK-CG were measured by Waters 1525 separations system (Waters S.A., Montreux-Chailly, Switzerland) with separation achieved on a series of columns composed of TSKgel G-6000PWxl (13 μM, 7.8 mm × 300 mm) and TSKgel G4000PWxl (10 μM, 7.8 mm × 300 mm) [25]. The separation system was equipped with a Shodex RI-101 RI detector (Showa Denko Europe GmbH, Germany) and performed at 35 °C. The columns were all held at 50 °C; injection volume was 100 μL. The mobile phase was $KH_2PO_4$ solution (0.02 mol/L, pH 6.0) and flow rate was set at 0.6 mL/min. Dextran was used as the standard with a concentration of 1.0 mg/mL.

## 2.3. FTIR Spectroscopy and $^{13}$C-NMR of K-CG, LK-CG

FTIR characterization of CG, LK-CG was conducted according to the previously reported method, with minor modifications [26]. Briefly, samples of powdered CG, LK-CG (10 mg each) were crushed in a mortar and pestle. The crushed sample was mixed with potassium bromide, dried, and compressed to pastilles. NiocetlMgana750 infrared spectrophotometer (Thermo Fisher, MA, USA) was used to obtain spectra in the wavelength range of 4000–500 cm$^{-1}$ with a resolution of 4 cm$^{-1}$ as baseline.

NMR analysis was performed by using an AVANCE DRX-200 MHz instrument (Bruker, Billerica, MA, USA). Sample (5.0 mg) of K-CG and LK-CG was mixed with 0.7 mL of $D_2O$ containing 1.0 mM of sodium trimethylsilylpropanesulfonate (DSS) as internal standard and 20 mM $Na_2HPO_4$ buffer [27]. $^{13}$C-NMR analysis was performed in 30,000 scans and the spectra were obtained at 35 °C probe temperature with 90° pulses and relaxation delay of 5 s.

## 2.4. Preparation of LK-CG-TP Microspheres

LK-CG-TP microspheres were prepared according to the method described by Ellis and Jacquier with minor modification [28]. LK-CG and TP (0.5 g) were suspended in distilled water, stirred, and held at 60 °C for 20 min to completely dissolve the solutes (100 mL). Then the solution (60 °C) was mixed with 500 mL rapeseed oil solution (60 °C) containing 1% Span80 (*w/v*) and 3% *n*-butanol (*w/v*) under magnetic stirring. The mixture was homogenized at 24,000 rpm in a high-speed homogenizer for 4 min. The concentrations of LK-CG, TP, rapeseed oil, Span80, and *n*-butanol were optimized in the preliminary experiments (Tables S1–S5). After homogenization, the emulsion was placed in iced water, thus dropping the temperature to approximately 10 °C, and maintained for 30 min [28]. Then, the oil phase was removed from the microspheres after 10 min of centrifugation (1500 rpm). The separated microspheres were further hardened and washed extensively with a 100 mM KCl solution, then freeze-dried to obtain LK-CG-TP microspheres.

## 2.5. Characterization Techniques

### 2.5.1. SEM Analysis of LK-CG-TP Powder

Samples of freeze-dried LK-CG-TP powder were attached to stubs using a two-sided adhesive tape, then coated with a gold layer (50 nm) using a sputter coater before observation [26]. Scanning electronic microscopy (SEM) analysis was performed using a Hitachi S4800 microscope (Hitachi High-Technologies, Tokyo, Japan) at an accelerating voltage of 3.0 kV.

### 2.5.2. TG Analysis and DSC Analysis

TG analysis was performed according to the previously reported method with some modifications [29]. Powdered samples of TP, LK-CG, and LK-CG-TP (10 mg each) were scanned using a Q5000 thermogravimetric analyzer (TA Instruments, New Castle, DE, USA). The analysis was performed in the temperature range 20–500 °C with a heating rate of 10 °C/min, under a controlled $N_2$ gas atmosphere with a flow rate 30 mL/min.

Differential scanning calorimetry (DSC) analysis was performed to thermally evaluate the LK-CG-TP with a DSC Q2000 instrument (TA Instruments, New Castle, DE, USA). Measurements were carried out under a controlled $N_2$ gas atmosphere with a flow rate 30 mL/min. The samples (5 mg) were placed in DSC pans and heated from −70 to 200 °C at the rate of 10 °C/min to obtain the DSC calibration [29].

### 2.5.3. In Vitro Release Study

In-vitro release study was performed using simulated gastric digestive fluid as described by Xie et al. [29]. For simulated gastric fluid (SGF) preparation, pepsin (1.0 g) and NaCl (0.2 g) were dissolved in water and 1 M hydrochloric acid solution was added to adjust the pH value (0.9–1.2) with a final volume of 100 mL. The prepared solution was centrifuged at 5000 rpm for 10 min, and the supernatant used as SGF. The powdered LK-CG-TP was added to the SGF solution with a water bath at 37 °C. Centrifugation was performed subsequently after 0 h, 0.5 h, 1.0 h, 1.5 h, 2.0 h, 2.5 h, and 3 h of digestion. The supernatant was obtained and was used to determine the content of EGCG by HPLC.

HPLC (Shimazu LC-20AT, Japan) with a PDA detector (SPD-M20A) and a $C_{18}$ column (4.6 mm × 250 mm, 5 μM, Agilent ZORBAX Eclipse XDB-$C_{18}$) was used for the qualitative and quantitative analysis of EGCG under 226 nm. The supernatant sample (20 μL) was eluted with a mobile phase, comprising 15% acetonitrile and 85% water with 0.1% acetic acid for 30 min. The elution flow rate was set at 1.0 mL/min and EGCG standard was used to construct a calibration curve by HPLC analysis.

### 2.5.4. DPPH Radical Scavenging Activity of TP and LK-CG-TP

TP and LK-CG-TP samples were dissolved in water (60 °C) with a final TP concentration of 10, 20, 30, 40, 50, and 60 μg/mL, respectively. A 5 mL sample solution and an equal amount of 0.1 mM DPPH ethanol solution or ethanol (used as blank) were fully mixed and placed in the dark for 1 h. The absorbance values of the sample solution in 0.1 mM DPPH and in ethanol were obtained at 538 nm as Asample and Ablank, respectively. The absorbance of DPPH in the equivalent phosphate buffer solution (Acontrol) was also measured. The free radical scavenging rate was calculated by equation of [1 − (Asample − Ablank)/Acontrol] × 100% [30].

### *2.6. Statistical Analysis*

All the experiments were carried out in triplicates and reported as mean ± standard deviation. Data was analyzed using a one-way analysis of variance (ANOVA) test. Moreover, all the graphs were processed with Origin 9.0 (OriginLab Co. Ltd., Northampton, MA, USA).

## 3. Results

### *3.1. Preparation and Characterization of LK-CG*

As shown in Table 1, the Mw and Mn of obtained LK-CG were $1.3 \times 10^4$ and $1.1 \times 10^4$, which were much lower than those of K-CG ($23 \times 10^4$ Da and $14 \times 10^4$ Da, respectively). In addition, the polymer dispersity index (PDI) of LK-CG was sharply decreased from 16.2165 to 1.13781 after degradation (Table 1).

**Table 1.** The Mw, Mn, PDI of K-CG and LK-CG.

| Sample | Mw (Da) | Mn (Da) | PDI |
| --- | --- | --- | --- |
| K-CG | 233,501.3 | 1,439,993.1 | 16.2165 |
| LK-CG | 13,009.5 | 11,433.8 | 1.13781 |

Mw: Mass-average molecular weight. Mn: Number-average molecular weight, PDI: polymer dispersity index. K-CG: Kappa–Carrageenan. LK-CG: lower molecular weight Kappa–Carrageenan.

The FTIR analysis of K-CG and LK-CG revealed a similar spectrum as shown in Figure 1, indicating that the functional groups and main molecular framework are retained after depolymerization. The band between 3600 and 3200 cm$^{-1}$ represents the stretching of the hydroxyl groups. The peaks at around 1220 cm$^{-1}$, 920 cm$^{-1}$, and 840 cm$^{-1}$ were attributed to symmetric stretching vibration of O–S–O, C–O–C, and C–O–S, respectively (Figure 1).

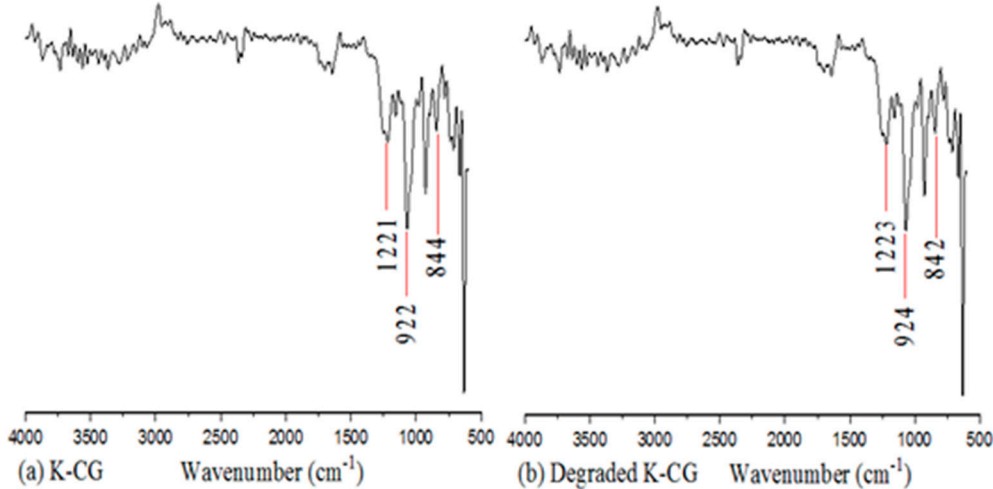

**Figure 1.** The infrared spectra of samples of K_CG (**a**) and after degraded K_CG (**b**).

The $^{13}$C-NMR analysis of the LK-CG provides insights on the structural changes caused by depolymerization treatment (Figure 2). The results indicated that the peaks appeared at 104.8 ppm; they were assigned to C-1 of 3-linked β-D-galactopyranosyl residue linked to→4-α-L-galactopyranosyl unit sulfated at position O-3 [31]. The peaks at 94.3, 76.5, and 68.9 ppm were found to correspond to α-1,4-linked-D-galactose [32]. The oxygenated carbon peak (79.2 ppm) appeared in the K-CG sample, but was not observed in the LK-CG sample, which may be due to degradation treatment [27]. As reported by Aji Prasetyaningrum [33], infrared spectroscopy (IR) can be used to determine the differences of sulphate groups between the degraded products and original ones. The obtained FT-IR spectra revealed that there are no significant changes in the functional groups and chemical structure of K-CG after degradation (Figure 1). As shown in Figure 2, the $^{13}$C-NMR spectra of the LK-CG showed similar general peaks with the K-CG. These results suggested that most of the original structure is still present in the degraded product.

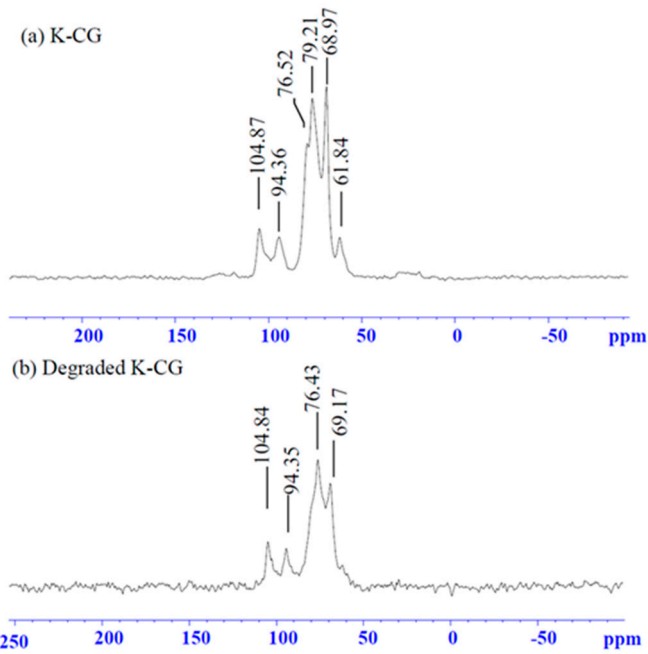

**Figure 2.** $^{13}$CNMR analysis of K_CG and degraded K_CG.

### 3.2. Characterization of LK-CG-TP Microspheres

### 3.2.1. Morphology of Microspheres

Morphological characterization and the surface charges of the microspheres were evaluated by scanning electron microscopy (SEM) and photocorrelation spectroscopy [34]. The SEM images of LK-GC-TP microspheres are shown in Figure 3. As shown in Figure 3A, microspheres overlap and most microspheres share spherical morphology with a mean diameter of 5–10 μM. In addition, the microspheres exhibited a porous, dense structure (Figure 3B).

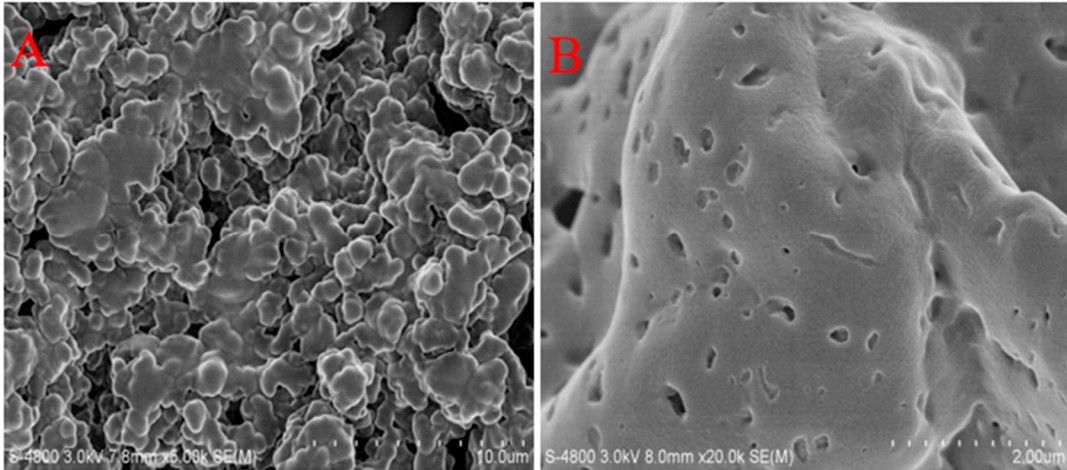

**Figure 3.** The SEM images of K-CG-TP mixtures ((**A**,**B**) respectively 5K×, 10K×).

### 3.2.2. The Thermal Stability of LK-CG-TP Microspheres

Thermo-gravimetric (TG) analysis of TP, LK-CG, and LK-CG-TP microspheres was carried out to analyze thermal stability of microspheres (Figure 4). The mass loss in TP and LK-CG powder started at 250 °C and 200 °C, respectively. As shown in Figure 4, 46% mass of K-CG powder was left after 500 °C. For microspheres, the first stage of the LK-CG-TP decomposition occurred within the scope of 50–100 °C, with a thermal weight loss of about 5.14% (Figure 4). The second phase ranged from 100 °C−190 °C, with a thermal weight

loss decrease of 11.36%. Due to the decomposition of TP and LK-CG, 80% of the mass of LK-CG-TP was lost in the temperature range of 200 °C to 420 °C and approximately 15% of the mass of the microspheres was left after 450 °C (Figure 4). The weight loss of LK-CG-TP was less as compared with TP and LK-CG in the temperature range of 20 °C to 350 °C, which indicated the protection ability of LK-CG on TP thermal degradation and high thermal stability of the LK-CG-TP complex. As compared to the TP and LK-CG samples, the sharply decreased weight loss of KL-CG-TP in the temperature range of 350 °C to 420 °C may be caused by the decomposition of the polysaccharide and TP structure, as well as the chemical reaction (such as Maillard reaction) triggered by the increase of temperature [35]. Similar results were also observed in the kappa-carrageenan-based anthocyanins nanocomplex [36].

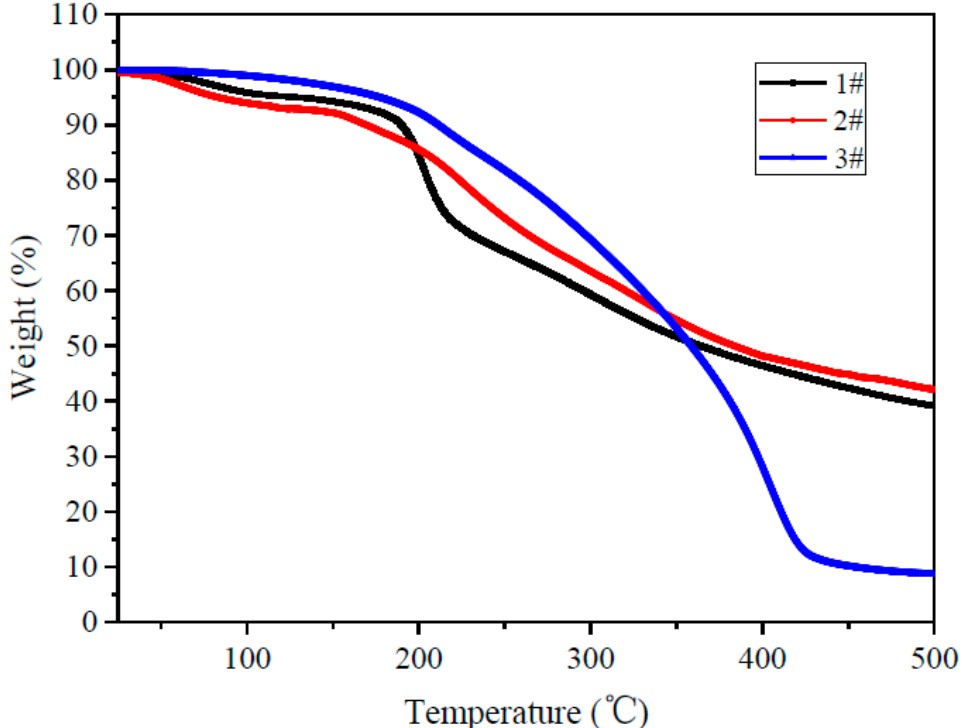

**Figure 4.** Thermogravimetric analysis of the microspheres (1 # TP, 2 # LK-CG, 3 # LK-CG-TP).

The calorimetric thermograms of LK-CG-TP microspheres are depicted as a plot of heat flow (W/g) versus temperature (°C), and are shown in Figure 5. The DSC thermograms of LK-CG-TP microspheres showed quite broad melting transition temperature range. The melting temperature of LK-CG-TP was found to be around 69 °C (Figure 5). The glass transition (Tg) of the microspheres was centered at 89.21 °C, which indicated the thermal stability of the microspheres under lower temperature. With the temperature raised (>Tg) in the subsequent heating scan, a sharp endothermic peak was formed due to possible LK-CG-TP degradation (Figure 5).

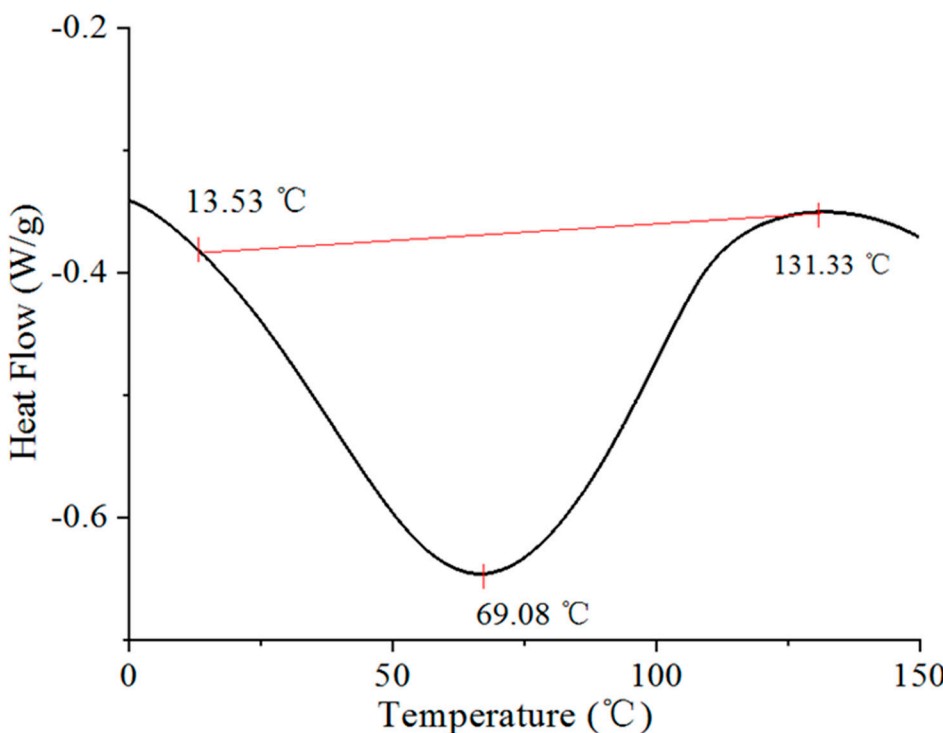

**Figure 5.** The differential scanning calorimetry analysis image of LK_CG_TP microspheres.

### 3.2.3. Release Profile of EGCG from LK-CG-TP Microsphere

The release profile of TP from the microspheres was investigated by HPLC analysis of the cumulative EGCG content in vitro using SGF solution, with free TPs used as control. The linear regression equation of EGCG was y = 8180.85 × −0.271 ($R^2$ = 0.9998), where y is the peak area and x is the EGCG concentration (mg/L), with the range of 20.00–100.00 mg/L. According to Figure 6, the content of EGCG in SGF solution decreased rapidly in the first 30 min for the TPs sample, and then slowed down in the subsequent 150 min. As compared to TPs, microspheres slowly released EGCG within tested 180 min, with the final release rate of 88.1% after digestion (Figure 6). These results revealed that LK-GC-TP complex microspheres exhibited potential good protective effects on EGCG (or TP) in the tested SGF solution.

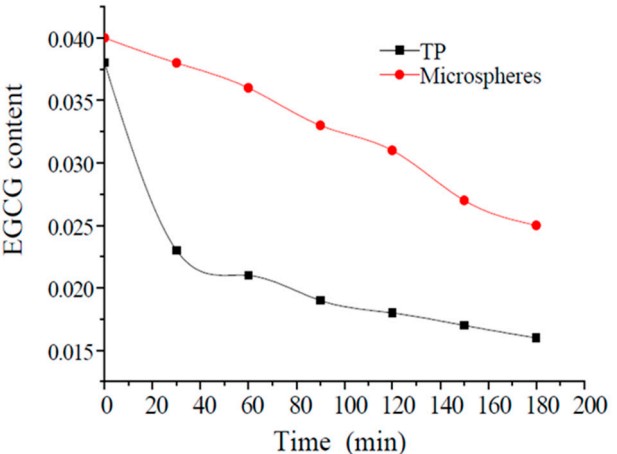

**Figure 6.** The variation of EGCG content during release process.

### 3.2.4. DPPH Scavenging Ratio of LK-CG-TP Microspheres and TP

DPPH scavenging ability is a widely used method to evaluate antioxidant activity in a relatively short time as compared with other methods. The DPPH free radical scavenging

rates of TP and LK-CG-TP microspheres are shown in Figure 7. The result revealed a positive correlation between DPPH scavenging rates and concentrations of the two tested samples (Figure 7). The antioxidant activity of LK-CG-TP solution increased from 13.28 ± 0.75% to 93.46 ± 0.51% with the TP concentration ranged from 10 µg/mL to 60 µg/mL. Notably, the DPPH scavenging rate of LK-CG-TP was higher than that of TP under the same Tp concentration. This may be because the scavenging activity of LK-CG is largely pronounced after K-CG degradation, as reported elsewhere [23], and would therefore result in the enhanced scavenging activity of LK-CG-TP microspheres.

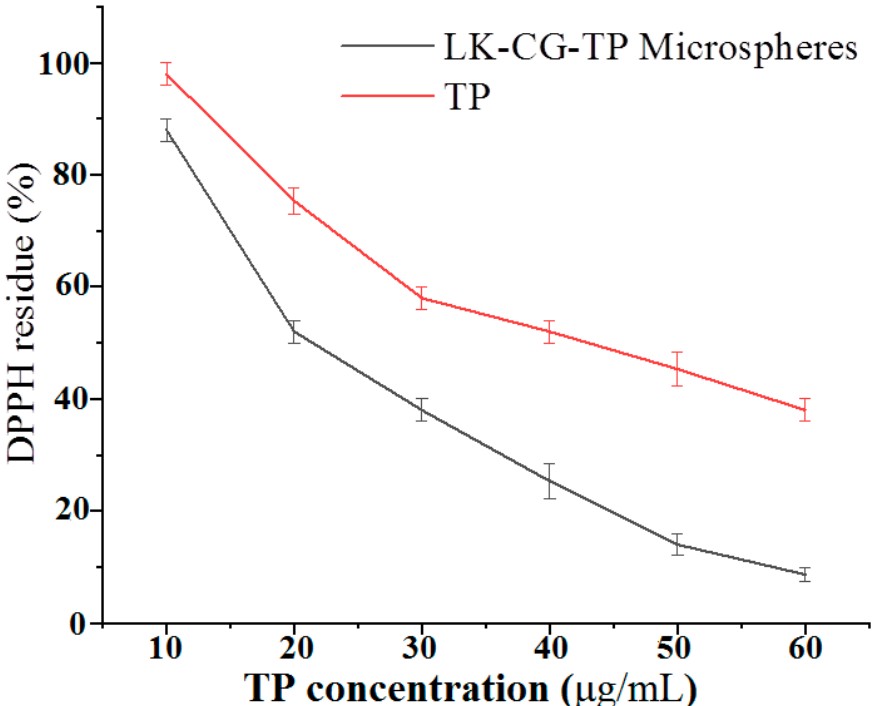

**Figure 7.** DPPH scavenging rate by concentration of TP and LK_CG_TP Microspheres.

## 4. Discussion

Various methods have been employed for the degradation of K-CG to produce LK-CG in previous approaches, which play important role in the structure, molecular weight, and bioactivity of the depolymerized products [23,24]. Sun et al. reported that the products of $H_2O_2$ hydrolysis revealed much higher antioxidant capacities as compared with other degradation methods. In this work, K-CG was degraded by $H_2O_2$ and further characterized by FT-IR and NMR technologies. The obtained LK-CG (~13.000 Da) would be a promising candidate for encapsulation of TP, as De Lima Barizao showed that low molecular weights of K-CG (ranging from 2000 to 25,000) exhibited more biological properties and were revealed to be a more proper encapsulation material [37]. In addition, high compatibility and structure make K-CG a suitable wall material for drug's sustainable release [38], which suggested its potential capacity in TP encapsulation and further functional products application.

SEM images revealed that an average diameter of 5–10 µM of microspheres were obtained in the newly developed LK-CG-TP. A similar result was also observed in carrageenan microspheres containing allopurinol, which indicated that the microsphere size was small than 40 µM when carrageenan was used as a base [38]. More importantly, the high sulfate and *n*-acetyl residues in LK-CG result in a hydrophilic surface and an excellent colloidal stability for the LK-CG-TP (Figure 3) microspheres [39]. Polysaccharides and/or protein-formed nanoparticles have been successfully developed for enhancing the absorption and bioavailability of TP in recent years [2]. More recently, Tian et al. improved the stability of TP by application of xanthan gum and locust bean gum [40]. For delivery of bioactive

components, K-CG is an affordable polysaccharide used in encapsulation as compared to xanthan gum and other polymers [37]. Therefore, the degraded K-CG encapsulated TP (i.e., LK-CG-TP) was supposed to be a more feasible strategy for industry production owing to its high efficiency in TP protecting and relative low price in manufacturing.

## 5. Conclusions

The present work aimed to develop a microsphere structure with LK-CG to encapsulate TP for bioactive component delivery under precise conditions. The degraded K-CG with lower molecular weight (Mw = 13,009.5) exhibited good TP entrapping effect. The obtained microspheres revealed high thermal stability and were stable in simulated gastric fluid digestion as compared to the free TP. DPPH scavenging experiments showed that the antioxidant activity of TP increased after LK-CG encapsulation. These results indicated that the utilization of LK-CG for delivery of TP could enhance the stability and bioavailability of TP.

**Supplementary Materials:** The following are available online at https://www.mdpi.com/article/10.3390/pr9071240/s1, Table S1: The impact of LK-CG on particle size, loading rate and encapsulation rate, Table S2: The impact of LK-CG to TP ratio on particle size, loading rate and encapsulation rate, Table S3: The impact of water to oil ratio on particle size, loading rate and encapsulation rate, Table S4: The impact of Span80 concentration on particle size, loading rate and encapsulation rate, Table S5: The impact of n-butanol concentration on particle size, loading rate and encapsulation rate.

**Author Contributions:** T.F.: Conceptualization, methodology, investigation, data curation; K.W.: investigation, writing original draft, formal analysis, data curation; J.X.: writing—review and editing, supervision, funding acquisition; Z.H.: reviewing and editing; X.Z.: conducting experiment. All authors have read and agreed to the published version of the manuscript.

**Funding:** This research received no external funding.

**Institutional Review Board Statement:** Not applicable.

**Informed Consent Statement:** Not applicable.

**Acknowledgments:** The technical assistance of Ling-Yun Yao (Shanghai Institute of Technology, Shanghai, China) is gratefully acknowledged.

**Conflicts of Interest:** The authors declare no conflict of interest.

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
