# Peer review of "Low Molecular Weight Kappa-Carrageenan Based Microspheres for Enhancing Stability and Bioavailability of Tea Polyphenols"

_processes, doi:10.3390/pr9071240_

Round 1

Reviewer 1 Report

Overall the manuscript is interesting, has good readability, and features essential sets of data for characterizing microspheres. The study is done very comprehensively with the application of several techniques, from the characterization of the microspheres to the release of tea polyphenols. 
Carrageenans are widely used in various drug delivery systems, but the use of modified carrageenans is still under investigation and this study would contribute to the development of microspheres with adequate stability and bioavailability properties. This is an interesting, comprehensive and well designed paper, and I think, it deserves publication after a minor revision. 

It seems to me that it is necessary to attach as annexes, the results of the preliminary experiments, in order to substantiate all the work.
In addition, it seems appropriate to argue the advantages of carrageenan over other polymers in the development of microspheres, being appropriate to add more information in the discussion section.

Author Response

Responds to the reviewer’s comments:

Reviewer #1: “It seems to me that it is necessary to attach as annexes, the results of the preliminary experiments, in order to substantiate all the work”.

Response:

  Thanks for the reviewer’s comments. We have attached the preliminary experiment results as supplementary materials.

Reviewer #1: “In addition, it seems appropriate to argue the advantages of carrageenan over other polymers in the development of microspheres, being appropriate to add more information in the discussion section.”

Response:

  Thanks for the reviewer’s comments. We have discussed the advantages of carrageenan over other polymers in discussion part (Line 296-299).

CG is an economic polymer than xanthan gum and other polymers [37]. In addition, CG is a more compatible and mild gelation polymer due to its structural strength [38]. Nowadays, CG has been widely utilized as wall material in various formulations for sustainable release [36].  

Reviewer 2 Report

The manuscript 'Low molecular weight kappa-carrageenan based microspheres for enhancing stability and bioavailability of tea polyphenols' demonstrate an efficient encapsulation strategy to increase the bioavailability of tea polyphenols. The manuscript is well organized and ideas have been conveyed with clarity. However, I would like the authors to address the following points.

  1. To begin with, In line 72 authors mention that K-CG has a higher electronegativity than alginate. As per the formal definition, electronegativity is a property of an atom to attract shared electrons to itself. the term 'electronegativity' can't be used for the molecules.
  2.  In line 273, the authors claim that the demonstrated materials show a regular spherical morphology with the help of an SEM image. However, In SEM image  3A, such regular morphology is not visible. 
  3.  In figure 4, the thermogravimetric data is presented.  after 400 degrees celsius, the remaining weight of the sample  LK-CG-TP  is around 10%, which is relatively lower compared to other samples. I would like the authors to have a discussion around this degradation behaviour. 

After addressing these comments, the manuscript should be considered for publication

Author Response

Reviewer #2:To begin with, in line 72 authors mention that K-CG has a higher electronegativity than alginate. As per the formal definition, electronegativity is a property of an atom to attract shared electrons to itself. the term 'electronegativity' can't be used for the molecules.

Response:

  Thanks for the reviewer’s comments. This term 'electronegativity' that inappropriate for molecules and was replaced by compatibility.

Reviewer #2:In line 273, the authors claim that the demonstrated materials show a regular spherical morphology with the help of an SEM image. However, In SEM image 3A, such regular morphology is not visible.

Response:

  Thanks for the reviewer’s comments. We have changed the inappropriate description and replaced Figure 3A in the revised MS (Line 230-232).

As shown in Figure 3A, microspheres are overlapped and most microspheres share spherical morphology and with a mean diameter of 5-10 μm.

Reviewer #2:In figure 4, the thermogravimetric data is presented.  after 400 degrees celsius, the remaining weight of the sample LK-CG-TP is around 10%, which is relatively lower compared to other samples. I would like the authors to have a discussion around this degradation behavior. 

Response:

  Thanks for the reviewer’s comments. We have added some discussion in the revised MS (Line 248-253).

As compared to TP and LK-CG sample, sharply decreased weight loss of LK-CG-TP in the temperature range of 350 °C to 420 °C may cause by the decomposition of the polysaccharide and TP structure, as well as chemical reaction (such as Maillard reaction) triggered by the increase of temperature [35]. Similar results were also observed in the K-CG based anthocyanins nanocomplex [36]. 

Reviewer 3 Report

Review Processes

Manuscript title: Low molecular weight kappa-carrageenan based microspheres for enhancing stability an bioavailability of tea polyphenols

In general

This article shows interesting results for the encapsulation of tea polyphenols by degraded form of kappa carrageenan.  Anyhow I think there is only a results presentation without careful analysis and with un unsatisfactory level of “micro” discussion.

Introduction

In general well-constructed with minor language problems for example L42: gut instead of good, wright?L.64 Latin names should be in italics.

Material and methods

I am really surprised by the very low molecular weight for the obtained kappa carrageenan as it was usually reported on the level of 106 to 107Da (Karlsson and Sinhg 1999, KamiÅ„ska-Dwórznicka et al. 2015). My other concern is that degraded carrageenan could cause serious gastric problems an it was reported in previous study Delahunty et al. 1987, Haijin et al., 2003 and Tobacman et al., 2013. Worldwide many countries have food low against using carrageenans with molecular mass lower than 105Da. How do you try to explain use of this kind of K-CG with quite low molecular Weight even before degradation….

Results

In text and also in Table 1 – L. 192-199 there is a lack of unit, Da???

  1. 2015-2016 – I need to have more information to this results. Is this a specific effect for this type of hydrolysis? Was it reported previously? Remember that similar C-NMR spectra doesn’t mean that there was no changer in the numerous of sulphate groups that could affect the carrageenan properties….please discuss this results wider

Morphological characterization is poor and marginally treated…

The calorimetric thermograms – hypothesis from L. 249 – please discuss it with some other results

Discussion

Presented discussion is not the discussion – again about the results without clearly stated research hypotheses and without their discussion with published results

Conclusion

According to all my doubts I pointed above, try to re-define some of your thesis.

References

Are they prepared according to any requirements?

Author Response

Reviewer #3:This article shows interesting results for the encapsulation of tea polyphenols by degraded form of kappa carrageenan. Anyhow I think there is only a results presentation without careful analysis and with un unsatisfactory level of “micro” discussion.

Response: Thanks for the reviewer’s comments. We have added some description in the resubmitted version.

Such as: As shown in Figure 3A, microspheres are overlapped and most microspheres share spherical morphology and with a mean diameter of 5-10 μm. With the encapsulation of LK-CG, it is evident that tremendous spheres were formed. 

Reviewer #3:In general, well-constructed with minor language problems for example L42: gut instead of good, wright? L. 64 Latin names should be in italics.

Response: Done as suggested.

Reviewer #3: I am really surprised by the very low molecular weight for the obtained kappa carrageenan as it was usually reported on the level of 106 to 107Da (Karlsson and Sinhg 1999, KamiÅ„ska-Dwórznicka et al. 2015). My other concern is that degraded carrageenan could cause serious gastric problems and it was reported in previous study Delahunty et al. 1987, Haijin et al., 2003 and Tobacman et al., 2013. Worldwide many countries have food low against using carrageenans with molecular mass lower than 105Da. How do you try to explain use of this kind of K-CG with quite low molecular Weight even before degradation.

Response: The molecular weight (Mw) for K-CG usually ranged from 106 to 107Da as reported previously, which depends on the materials and extraction methods. For example, Relleve & Abad (2015) reported the extracts consisted of low Mw fragments with an average Mw ranging from 2300 Da to 5000 Da. Tecson et al. (2021) reported that long treatment time (180 mins) afforded a degraded K-CG with average Mw of 41,864 Da, which is a 96.33% reduction from the raw sample with initial Mw of 1,139,927 Da. Abad et al. (2016) observed that K-CG with Mw<6000 Da have good plant growth promoter effect and GPC analyses indicated at least three LMW fragments with an Mw<2 kDa.

In this work, the low Mw K-CG-TP complex was developed mainly for skin care and/or cosmetic products. Therefore, the potential gastric problems can be ignored in future utilization.

Relleve L. & Abad L. (2015). Characterization and antioxidant properties of alcoholic extracts from gamma irradiated κ-carrageenan. Radiation Physics and Chemistry, 112, 40-48.

Lucille V.Abad, Fernando B.Aurigue, Lorna S.Relleve, Djowel Recto V.Montefalcon, Girlie Eunice P.Lopez. (2016). Characterization of low molecular weight fragments from gamma irradiated κ-carrageenan used as plant growth promoter. Radiation Physics and Chemistry, 118, 75-80.

Tecson, M.G.,Abad, L.V.,Ebajo Jr., V.D., Camacho,D.H. (2021). Ultrasound-assisted depolymerization of kappa-carrageenan and characterization of degradation product. Ultrasonics Sonochemistry, 73, 105540.

Reviewer #3: Results, in text and also in Table 1 – L. 192-199 there is a lack of unit, Da???

Response: Done as suggested.

Reviewer #3: 215-216 – I need to have more information to this results. Is this a specific effect for this type of hydrolysis? Was it reported previously?

Response: Yes. Actually, the hydrolysis of K-CG has been widely investigated previously as the literature below.

Aji Prasetyaningrum, B. J., Ratnawati Ratnawati, Effect of ozonation process on physicochemical and rheological properties of k-carraageenan. Scientific Study & Research 2017, 18(1), 109-118.

Reviewer #3: Remember that similar C-NMR spectra doesn’t mean that there was no changer in the numerous of sulphate groups that could affect the carrageenan properties….please discuss this results wider.

Response: Thanks for the reviewer’s comments. We have added some discussion in the resubmitted MS (Line 214-218).

As reported by Aji Prasetyaningrum [33], infrared spectroscopy (IR) can be used to de-termine the differences of sulphate groups between the degraded products and original ones. The obtained FT-IR spectra revealed that there are no significant changes in the functional groups and chemical structure of K-CG after degradation (Figure 1).

Reviewer #3: Morphological characterization is poor and marginally treated…

Response: Done as suggestion.

Reviewer #3: The calorimetric thermograms – hypothesis from L. 249 – please discuss it with some other results.

Response: Done as suggested.

Reviewer #3: Presented discussion is not the discussion – again about the results without clearly stated research hypotheses and without their discussion with published results.

Response: Thanks for the reviewer’s comments. We have strengthened the results and discussion in the revised MS.

For example: With the encapsulation of LK-CG, it is evident that tremendous spheres were formed. What’s more, stability and bioavailability of tea polyphenols with the protection of wall material. However, microspheres are not evenly dispersed but overlapped and mixed, which will be However, microspheres are not evenly dispersed but overlapped, which will be further investigated.

Round 2

Reviewer 3 Report

Some points of my concerns were corrected, however the discussion part was only slightly corrected ant the added parts are more like from the introduction not from the discussion. Authors should explain their results comparing to results form other researchers and the didnt give examples in here only basing on the general sentences its not the discussion for me.

Author Response

Responds to the reviewer’s comments:

Reviewer #3: “Some points of my concerns were corrected, however the discussion part was only slightly corrected ant the added parts are more like from the introduction not from the discussion. Authors should explain their results comparing to results form other researchers and the didnt give examples in here only basing on the general sentences its not the discussion for me.”

Response:

  Thanks for the reviewer’s comments. We have intensified the discussion in the revised MS, including the degradation of K-CG and LK-CG formed microspheres (Line 289-314).

Various methods have been employed for degradation of K-CG to afford LK-CG in the previous approaches, which play important role in the structure, molecular weight and bioactivity of the depolymerized products [23, 24]. Sun et al. reported that the products of H2O2 hydrolysis revealed much higher antioxidant capacities as compared with other degradation methods. In this work, K-CG was degraded by H2O2 and further characterized by FT-IR and NMR technologies. The obtained LK-CG (~13000 Da) would be a promising candidate for encapsulating of TP, as De Lima Barizao described that low molecular weight of K-CG (ranged from 2000 to 25,000) exhibited more biological properties and revealed to be a more proper encapsulation material [39]. In addition, high compatibility and structural makes K-CG a suitable wall material for drug’s sustainable release [36, 38], which suggested its potential capacity in TP encapsulation and further functional products application.

SEM images revealed an average diameter of 5-10 μm of microspheres were obtained in the newly developed LK-CG-TP. Similar result was also observed in carrageenan microspheres containing allopurinol, which indicated that the microsphere size was small than 40 μm when carrageenan was used as a base [38]. More importantly, the rich contained sulfate and N-acetyl residues in LK-CG would result in a hydrophilic surface and an excellent colloidal stability for the LK-CG-TP (Figure 3) microspheres [39]. Polysaccharides and/or proteins formed nanoparticles have been successfully developed for enhancing the absorption and bioavailability of TP in a recent years [2]. More recently, Tian et al. improved the stability of TP by application of xanthan gum and locust bean gum [40]. For delivery of bioactive components, K-CG is an economic polysaccharide used in encapsulation as compared to xanthan gum and other polymers [37]. Therefore, the degraded K-CG encapsulated TP (i.e., LK-CG-TP) was supposed to be a more feasible strategy for industry production owing to its high efficiency in TP protecting and relative low price in manufacturing.

[2] Massounga Bora, A. F.; Ma, S.; Li, X.; Liu, L., Application of microencapsulation for the safe delivery of green tea polyphenols in food systems: Review and recent advances. Food Res Int 2018, 105, 241-249.

[23] Sun, Y.; Yang, B.; Wu, Y.; Liu, Y.; Gu, X.; Zhang, H.; Wang, C.; Cao, H.; Huang, L.; Wang, Z., Structural characterization and antioxidant activities of kappa-carrageenan oligosaccharides degraded by different methods. Food Chem 2015, 178, 311-8.

[24] Wu, S. J., Degradation of kappa-carrageenan by hydrolysis with commercial alpha-amylase. Carbohydr Polym 2012, 89, (2), 394-6.

[36] Hadiyanto, H.; Christwardana, M.; Suzery, M.; Sutanto, H.; Nilamsari, A. M.; Yunanda, A., Effects of Carrageenan and Chitosan as Coating Materials on the Thermal Degradation of Microencapsulated Phycocyanin from Spirulina sp. International Journal of Food Engineering 2019, 15, (5-6).

[37] Pourashouri, P.; Shabanpour, B.; Heydari, S.; Raeisi, S., Encapsulation of fish oil by carrageenan and gum tragacanth as wall materials and its application to the enrichment of chicken nuggets. Lwt 2021, 137.

[38] Ashe, S.; Behera, S.; Dash, P.; Nayak, D.; Nayak, B., Gelatin carrageenan sericin hydrogel composites improve cell viability of cryopreserved SaOS-2 cells. Int J Biol Macromol 2020, 154, 606-620.

[39] de Lima Barizao, C.; Crepaldi, M. I.; Junior, O. O. S.; de Oliveira, A. C.; Martins, A. F.; Garcia, P. S.; Bonafe, E. G., Biodegradable films based on commercial kappa-carrageenan and cassava starch to achieve low production costs. Int J Biol Macromol 2020, 165, (Pt A), 582-590.

[40] Tomoda, K.; Asahiyama, M.; Ohtsuki, E.; Nakajima, T.; Terada, H.; Kanebako, M.; Inagi, T.; Makino, K., Preparation and properties of carrageenan microspheres containing allopurinol and local anesthetic agents for the treatment of oral mucositis. Colloids Surf B Biointerfaces 2009, 71, (1), 27-35.